# Locating Partial Discharges in Power Transformers with Convolutional Iterative Filtering [note 1]

**DOI:** 10.3390/s23041789

**Published:** 2023-02-05

**Authors:** Jonathan Wang, Kesheng Wu, Alex Sim, Seongwook Hwangbo

**Affiliations:** 1Lawrence Berkeley National Laboratory, Berkeley, CA 94720, USA; 2Vitzrotech Co., Ltd., Seoul 425833, Republic of Korea

**Keywords:** partial discharges, source location, UHF measurements, time of arrival estimation, waveform analysis, FDTD methods, nonlinear wave propagation

## Abstract

The most common source of transformer failure is in the insulation, and the most prevalent warning signal for insulation weakness is partial discharge (PD). Locating the positions of these partial discharges would help repair the transformer to prevent failures. This work investigates algorithms that could be deployed to locate the position of a PD event using data from ultra-high frequency (UHF) sensors inside the transformer. These algorithms typically proceed in two steps: first determining the signal arrival time, and then locating the position based on time differences. This paper reviews available methods for each task and then propose new algorithms: a convolutional iterative filter with thresholding (CIFT) to determine the signal arrival time and a reference table of travel times to resolve the source location. The effectiveness of these algorithms are tested with a set of laboratory-triggered PD events and two sets of simulated PD events inside transformers in production use. Tests show the new approach provides more accurate locations than the best-known data analysis algorithms, and the difference is particularly large, 3.7X, when the signal sources are far from sensors.

## 1. Introduction

Power transformers are among the most expensive components of electric power infrastructure. Although they are reliable, when one of them does fail, it has severe consequences; millions of users are affected each year, costing billions of dollars [1]. In addition, the grid operators also face significant penalties. To prevent such failures, power engineers have developed extensive online diagnostic systems [2,3,4]. With these sensors installed inside the transformers, effective data analysis is an active research topic [5,6,7]. This work reviews these state-of-the-art data analysis algorithms and refines two new techniques to more accurately locate the early warning signs of a common fault [7].

With some weakness in the insulation system, transformers become susceptible to failures [1]. One important sign of this insulation weakness is an internal arcing event known as partial discharge (PD) [8,9,10]. If the insulation involved in these discharges was not repaired or replaced, the insulation would be further damaged, leading to serious faults. Detecting and locating PD events allows one to take appropriate preventive actions [4,11,12,13]. Such preventive measures also protect other equipment connected to the transformers.

Many monitoring tools have been developed to assess the internal condition of transformers [8,11,12,14]. The best-known among them is dissolved gas analysis (DGA), which also detects internal electrical discharges. Even though DGA provides information about the nature and severity of the PD, it does not provide the location needed for remedial actions [3,15].

Other methods to locate PD include transformer winding modeling [4,16], acoustic methods [17,18], and ultra-high frequency (UHF) sensors [19,20,21]. Among these, UHF sensors are the most effective for locating PD positions [13,17,20,22]. However, processing the UHF sensor data is challenging for a number of reasons. For example, these UHF sensors produce a number of data records every nanosecond; processing data at this high rate requires careful planning and efficient processing algorithms. Furthermore, when a signal first arrives at a sensor, it is buried among the background noise. Extracting the signal arrival time from these noisy data records is an active research topic [5,18,19,23]. One key contribution of this work is a refinement of the Savitzky–Golay filter [7] to produce a more accurate approach for determining the arrival time (in Section 3).

After determining the arrival time, the time differences are used to deduce the location of the PD source [19,22,23,24,25]. However, because a transformer has a large amount of metal inside, the electromagnetic waves travel in complex paths, causing the popular triangulation methods to yield inaccurate locations [4,5,6]. To accurately locate the PD source, this work uses a table lookup approach [7]. This reference table of travel time is generated from detailed finite-difference time-domain (FDTD) simulations [4,26,27]. By matching observed time differences with the computed values in the reference table, the PD source location can be determined to a high accuracy, as shown in Section 4 and Section 5. The second contribution of this works is the use of a kd-tree to accelerate the lookup operation on reference table (details in Section 4).

To validate the new timing method and localization method, their predictions are compared with the known locations of PD events. These PD events consist of lab-generated events as well as simulated events. All the PD events are created from actual transformers produced by a well-known manufacturer. This evaluation is more systematic and makes use of more transformers than previously published studies [7].

In the remainder of this paper, a brief review of PD location techniques is presented in Section 2, the new timing technique is given in Section 3, and the algorithm for localization is described in Section 4. The experimental results are discussed in Section 5. A short conclusion and plan for future work is offered in Section 6.

## 2. Background and Related Work

This section contains a brief description of well-known analysis algorithms for locating the PD source from UHF sensor data. It starts with a short introduction of the data, then provides an overview of common approaches, and finally gives the two most successful techniques so far. More extensive reviews, especially about the power engineering aspect, are available from Mondal and colleagues [3,19].

### 2.1. UHF Data

This work focuses on locating the position of a PD event using UHF sensor data [4,28]. All data sets come from the same type of UHF sensor that records a voltage value every 0.4 ns. Four such sensors are used together inside a transformer, and these four streams of data are referred as four channels. The sensors are continuously reading the voltage values to memory, but the output mechanism is only triggered when one of the four channels produces a voltage reading above a preset threshold. Once triggered, a PD event is declared, and each channel outputs 500 values before and 500 values after the trigger. Altogether, the data for a PD event includes four series of 1000 voltage values each. This triggering and recording procedure is widely reported in the power engineering literature [4,17,18,19,21].

The primary test transformer, which we will call KE20, was set up with two distinct PD locations, PD1 and PD2 (more details in Section 4.3). Using this transformer, partial discharge events are induced from these two locations one at a time. The location PD1 is closer to the sensors than PD2, therefore, the signals recorded for events from PD1 generally have higher signal-to-noise ratios (SNR) than those events from PD2. The test data set contains 22 events from PD1 and 23 events from PD2. Figure 1 shows a sample of the sensor data. The sample in Figure 1a has relatively high signal-to-noise ratio (SNR), thus it is simpler to determine the arrival of the signal. This data is from channel 1 (CH1) of an event originated from PD1. The signal shown in Figure 1b is not much stronger than noise. This data is from channel 4 (CH4) of a PD2 event, where the signal has to propagate around the internal metal structures.

This work involves three different transformers of varying sizes. Much of the detailed study on how to construct the new algorithms is performed on transformer KE20. Once the analysis algorithms are calibrated, further validation is performed with two other transformers named DL23 and TL19.

### 2.2. General Approach for Locating PD

With a set of effective sensors, the source location of the signal could be determined from arrival time differences [4,19,24,25]. This approach is used in many applications. To determine the location accurately, one needs to determine the arrival time very precisely [12]. However, when the signal first arrives at a sensor it might be weaker than the background noise, and it could only be confidently identified when the signal strength is significantly stronger than noise. To determine when a signal is noticeably stronger than noise, a number of different smoothing techniques have been proposed [4,5,12,19], most of which are based on intuition informed by the physics of the signals involved. This work takes the signal processing approach to explore how to identify signal arrival when the signal is not much stronger than the noise.

After determining the signal arrival time, the next step is to perform the triangulation based on the time differences, the most commonly used implementation of which is known as time difference of arrival (TDOA) [23,29,30]. This approach assumes the signal travels in straight lines and at the same speed. Both of these assumptions are violated inside a transformer because the internal structure of a transformer is complicated, forcing most signals to travel in complex paths, and the media carrying the signal changes from location to location, causing the signal to travel at different speeds. Even though TDOA implementations contain sophisticated corrections to compensate for these inaccurate assumptions, exploring new localization algorithms without these assumptions could lead to more accurate answers.

### 2.3. Cumulative Energy Method

Based on published literature, the most effective method for identifying PD signal arrival time is the cumulative energy method [5,12,21], which is the reference method for evaluating newly developed algorithms. In the cumulative energy method, the voltage measurement $s_{i}$ is converted to cumulative energy by $C\left(t\right)=\sum_{i=1}^{t}s_{i}^{2}$, where $s_{i}$ is the voltage of the signal at time *i* and C(t) is the cumulative energy at time *t*. The intuition behind this approach is that the knee in the cumulative energy curve is where the energy from the incoming signal overtakes the background noise and therefore could represent the arrival of the signal. There are different ways to locate this “knee point” with a computer algorithm [31]. Next we describe a popular method.

Figure 2a illustrates a common way of selecting a knee point from the cumulative energy. In this procedure, the knee is the point furthest from a vector connecting the start of the cumulative energy to the end (labeled b in Figure 2a) and $b^{⊥}$ is the line representing the distance from b to a point on the cumulative energy curve. The x-coordinate (i.e., time) of the knee on the cumulative energy curve is the considered the arrival time of the PD signal.

### 2.4. Energy Criterion

Another successful method for determining signal arrival time is known as the energy criterion [12,18]. It avoids the complex procedure for finding the knee point of the cumulative energy by adding a negative bias. The energy criterion is given by
$$E\left(t\right)=\sum_{i=1}^{t}\left(s_{i}^{2}-t\delta \right),\;\delta =\frac{\sum_{i=1}^{N}s_{i}^{2}}{N}$$

The energy criterion method uses a negative trend δ to separate the signal from the noise. This trend is a function of the total energy of the signal and the length of the signal. The point with the minimum energy E(t) is considered as indicating the arrival of signal. This point is where the energy signal outpaces the negative trend. Figure 2b shows how the energy criterion is used to identify the arrival of the signal.

## 3. Arrival Time

As described in the previous section, the process of using UHF signals to determine the PD location is broken up into two steps: identifying the arrival time of the signal from the sensor data and localizing the partial discharge using the signal timings. There are challenges in both steps. This section focuses on addressing the challenges from the first step of pinpointing the arrival time of the signal from the sensor data. The discussion starts with a review of common algorithms and ends with proposing a composite method named convolutional iterative filter with thresholding (CIFT).

### 3.1. Thresholding

Following the description of the data collection procedure in Section 2, the data points at the beginning of a time series could be regarded as noise. A straightforward strategy is to set a threshold based on the mean and standard deviation of the noise, and declare the first point above the threshold as the signal arrival time.

When the signal first arrives, it might be much weaker than the threshold and would take some time to grow and reach the threshold. If the shape of the signal is known, it could be possible to deduce the actual arrival time. For example, assuming the measured voltage is the combination of a small number of plain sine waves and the leading edge of signal is a sine wave, then it is possible to compute the time needed for the signal to grow in strength to reach the threshold. This creates a simple correction mechanism for the basic thresholding technique to detecting signal arrival. In later tests shown in Table 2, this correction mechanism assumes the signal to be a sine wave with the dominant frequency that remain the same in a small time window.

### 3.2. Data Spread

Instead of relying on the mean and standard deviation to describe the noise, it is possible to set threshold using another concept called *spread*. The spread of a window is the difference between the maximum and minimum points in the window. In this approach, the maximum spread is computed from the noise, and the signal is declared as having arrived when the voltage goes outside of the noise spread. Note that this method is sensitive to spikes, especially in cases where signals are relatively weak.

### 3.3. Low-Pass Filter

From the published studies, the noise has much higher frequencies than the signal, so we could reduce the noise with a low-pass filter. However, because the frequencies of the signal are not well-separated from those of the noise, the low-pass filter could significantly reduce the signal strength. Tests shown that when SNR is low, the low-pass filter does not improve the SNR enough to define a good threshold. Figure 3 illustrates the magnitude reduction caused by the low-pass filter, and Figure 4a shows that the noise in the low SNR case is still very high.

### 3.4. Wavelength Comparison

A closely related idea in exploring the difference in the frequencies between signal and noise is to identify the frequency change to detect the arrival of the signal. However, this requires one to perform FFT on a fairly large number of data points, and therefore it could not provide a precise location of the arrival time.

### 3.5. Noise Cancellation

Since the data for each event only covers 400 ns of time, we might expect that the noise in such a short time duration could be considered as generated from a fixed source. In which case, the known noise (say, the first 400 data points of each channel) could be decomposed into a combination of simple wave forms, and we could subtract this combination of simple wave forms from the whole time series. This approach should cancel out the known noise components and improve SNR. Unfortunately, the noise frequencies change a lot even within a few nanoseconds (ns). Tests did not show a noticeable improvement to SNR.

### 3.6. Moving Average

To smooth over high-frequency noise, a common technique is the moving average. However, as with the low-pass filter, it is ineffective in the low SNR cases. Figure 4 shows that moving average is able to preserve height of the peaks better than the low-pass filter, but does not improve contrast between signal and noise.

### 3.7. Envelope

Instead of relying on frequency differences between signal and noise, there are a number of techniques attempting to trace the envelopingof the measured waves. Based on the literature, the upper envelope from a Hilbert transform is known to be effective in separating a PD signal from noise. Figure 5a shows the smoothed envelope over a strong signal. Notice that envelopes from different channels have different numbers of peaks, making it difficult to align signals to determine the signal arrival. In low SNR cases, the peaks of the signals are not significant different from those from the noise, as illustrated in Figure 5b, making it difficult to declare which is the signal.

### 3.8. Convolutional Filter

Another class of methods is a convolutional filter called the *Savitzky–Golay filter*, or SG filter for short [32]. In a related application, it allowed robust detection of signal from noisy data [33]. The SG filter smooths data by fitting a low-degree polynomial in a time window using the least squares.

Figure 6 shows that the SG filter improves SNR more than the low-pass filter. Even in the worst case (channel 4), the signal has doubled the amplitude of the signal using SG filter. The key shortcoming of SG filter is that it only identifies a time window where the signal is expected to have arrived. Additional work is needed to identify the precise arrival time within in the time window.

### 3.9. CIFT

The SG filter finds a time window where the signal appears for the first time. With this time window, another algorithm could identify the precise arrival time. Through some testing, we find the moving-average approach is the most effective for determining the precise arrival time within the time window. On the original observation, the moving average would reduce the amplitude of peaks too much, as illustrated in Figure 4b, making it difficult to find the arrival time. However, this moving-average approach is effective within a narrow window containing a single peak.

In summary, through extensive explorations, a two-stage algorithm is designed to more reliably determine the arrival of signals recorded by UHF sensors. This algorithm employs a convolutional filter to identify a candidate time window and then applies a standard method such as the moving average to more precisely pinpoint the arrival time. The resulting method is named convolutional iterative filter with thresholding (CIFT).

## 4. Localization

After determining the arrival time, the next task is to locate the source of the signals; this task is known as localization. The most common localization algorithms are variations of multilateration, which is the core of TDOA [23,30]. As an alternative to this approach, this section proposes a new method to lookup the location using the arrival time differences as the key to a reference table. This reference table is constructed through the finite-difference time-domain method [27] frequently used by power engineers to understand the wave propagation inside a transformer (see for example [4]).

### 4.1. Multilateration

Given four UHF sensors at known locations inside a transformer, the multilateration solves the following quadratic equations (i=1,2,3,4):$$\begin{array}{c} {\left(x-x_{i}\right)}^{2}+{\left(y-y_{i}\right)}^{2}+{\left(z-z_{i}\right)}^{2}-{\left(c_{e}\left(t-t_{i}\right)\right)}^{2}=0\;c_{e}=c/\sqrt{\epsilon_{r}} \end{array}$$
where *x*, *y*, and *z* are the coordinates of the PD source, and *t* is the corresponding time of the event, $c_{e}$ is the effective speed of light (c) traveling through the oil in the transformer, and $\epsilon_{r}$ is the relative permittivity of the oil (in many cases, $\epsilon_{r}\approx 1.7$). For i=1,2,3,4, $x_{i}$, $y_{i}$, and $z_{i}$ are the coordinates of the sensors, and $t_{i}$ is the time that the signal reaches sensor *i* computed using the above timing procedures. The sensor coordinates are known and fixed inside a transformer. In the later experiments, the variables *x*, *y*, *z*, and *t* are computed from the above equations using a derivative-free spectral residual method [34], which is an effective method to determine location from arrival time values. The key assumption in the above equations is that the signals travel from sources to the sensors in straight lines without obstruction; however, due the large amount of metal inside a transformer, this assumption is violated as illustrated next.

A heat map depicting the travel time from PD sources in a 2D plane is shown in Figure 7, where the sensor is at the upper left corner and the blue color represents an earlier arrival time. The blank spots are occupied by the metal structures. It is clear that the FDTD cell B is farther from the sensor than cell A, but its travel time to the sensor (blue) is shorter from cell A (green). There are other locations that are similarly further away from the sensor than their neighbors, but that have a bluer color, indicating a shorter travel time. The extensive metal structures inside a transformer force the electromagnetic waves to travel around them, which violates the straight line travel assumption behind the multilateration approach. Thus, it is worthwhile to explore localization approaches that do not depend on this assumption.

### 4.2. Table Lookup

The new localization algorithm utilizes a reference table to find the location where the travel time differences most closely matches with the observation. Key components of this algorithm, the reference table and the lookup procedure, are explained next.

Power engineers often use FDTD simulation to examine the electromagnetic waves traveling inside a transformer. This simulation can account for the impact of metal structures inside a transformer on travel time. The reference table is generated by simulating a PD event from each mesh point inside the transformer (that is not metal) and determining the arrival time at each of the sensors. To limit the computations need to generate this table, the mesh size is set to be 300 mm and time steps between 0.006 and 0.009 ns, which are small enough to model 3GHz UHF waves.

Conceptually, the lookup procedure goes through all records of the reference table to find the best match in the time differences. Since sensor 2 consistently has better SNR than other sensors in our experiments, the sensor 2 timing is used as the base for the time differences. Let $T_{i,j}$ be the observed time difference between sensors *i* and *j* computed from a timing procedure and $P_{i,j}^{\left(k\right)}$ be the computed time difference between sensors *i* and *j* for mesh point *k* of the FDTD mesh used to generate the reference table. Using sensor 2 as the base, the localized PD source is considered as point *k* that minimizes the following:$$\sqrt{\sum_{i\in \{1,3,4\}}{\left(T_{i,2}-P_{i,2}^{\left(k\right)}\right)}^{2}}.$$

In our software implementation, a kd-tree could be used to partition the $P_{i,j}^{\left(k\right)}$ table (based on [$P_{1,2}^{\left(k\right)}$, $P_{3,2}^{\left(k\right)}$, $P_{4,2}^{\left(k\right)}$]), so that many of the partitions would be pruned quickly given a set of [$T_{1,2}$, $T_{3,2}$, $T_{4,2}$].

Figure 8 shows the entirety of the PD localization procedure, consisting of finding the timing for each channel of the signal and then using those timings to lookup the nearest mesh point.

### 4.3. Consensus Location: Coping with Uncertainty in Lab-Triggered PDs

At PD1 and PD2, a dipole antenna with a standardized voltage source (IEC 60270) was used to generate PD events. Each antenna has 26 possible positions where a spark might appear. In the later discussion, these precise positions are referred to as PD coordinates. Since the triggering mechanism does not provide the precise coordinates, the computed locations could not be compared directly. However, if the locations are computed correctly, the computed locations should match with the two clusters represented by the two triggering antennas. Thus the computed locations are first clustered, where the cluster centers could be considered as the consensus locations of the triggering antennas.

A density-based spatial cluster algorithm named DBSCAN [35] is used to produce consensus locations. This clustering algorithm does not need users to specify the number of clusters as an input. It labels each derived source location as either a part of a cluster or as outliers. Additionally, any derived source location farther than 1000 mm from the nearest cluster is also counted as outliers. If the localization procedure were perfect, the clustering algorithm would identify two clusters around the two known antennas, with no outliers. The number of outliers could measure the quality of the localization procedure.

Figure 9 shows the workflow to compare the derived source locations against the known PD locations. Each PD event is used to compute a derived source location using its four channels. In the later evaluations, the main clusters identified by DBSCAN are compared with the known antenna locations, PD1 and PD2. The comparison is measured by *accuracy* and *total hits* defined as follows.

First, a tolerance range *d* is selected. Given *d*, a 2d diameter cube is drawn around a derived source location. Actual PD coordinates within that cube are within *d* distance of the derived source location in all dimensions, *x*, *y*, and *z*.

Let *P* denote the set of derived source locations, *Q* denote the points in the two main clusters, $E_{i}$ be the indicator of derived source location *i* containing a PD coordinate within its tolerance range, and $H_{i}$ be the number of PD coordinates within the tolerance range of location *i*.
$$\begin{array}{c} \mathrm{accuracy}=\frac{\sum_{i\in N}E_{i}}{\left|N\right|},\;\mathrm{total}\;\mathrm{hits}=\sum_{i\in P}H_{i} \end{array}$$

The ideal result would have no outliers, accuracy =1, and a large number of total hits.

## 5. Empirical Validation

To evaluate the effectiveness of the proposed PD location method, this numerical evaluation first works with the transformer KE20, which could be thought of as the training transformer, and then moves on with validation tests on two other transformers, DL23 and TL19. In the discussion of KE20, more details about localization methods and timing methods are provided.

### 5.1. Partial Discharge Localization Methods

This first set tests are to select a localization method to find the PD source after the arrival time has been computed. In this test, the well-known cumulative energy method is used to find the signal arrival time. The test compares multilateration with the FDTD table lookup method, using lab-generated PD from either PD1 or PD2 in KE20. The quality measures of accuracy and total hits defined earlier are computed with a tolerance range of 500 mm. Note that Zheng et al. [4] asserted that with a timing accuracy of 0.5 ns they could locate PD to 100 mm accuracy within the transformer winding. However, the same accuracy is not achievable in insulators. The error in arrival time could reach 10 ns in a case shown in Figure 10; and the combination of cumulative energy for timing and multilateration could locate a PD source within 500mm. The proposed CIFT and table-lookup combination could locate PD to about 300 mm accuracy.)

Figure 11 and Figure 12 show the positions of the actual and derived PDs. Figure 11 shows that the points determined by multilateration could be far from the actual PD coordinates. About half of the points are not even within 500 mm of the known PD locations. On the other hand, using the table lookup method on the same input data, there are no outliers, and all of the derived source locations are well within the 500 mm of the known PD locations, as shown in Figure 12. There are fewer than 22 blue squares in this figure because the table lookup produces identical locations in many cases.

Table 1 shows that the table lookup method has a much higher accuracy for both PD sources than multilateration. The table lookup method also computed PD locations without any outliers.

### 5.2. Signal Timing Methods

The last set of tests show that the table lookup approach is better than conventional multilateration for determining PD locations. Using this localization method, the next set of tests compares the performance the data-driven methods for identifying the signal arrival time described in Section 4. The comparisons use the same quality measures, accuracy and total hits. A summary of the results is given in Table 2.

From the previous section, we know the cumulative energy method could localize PD signals to within 500 mm of the known location; the next set of tests continue with this tolerance. As seen in Table 2, both cumulative energy and CIFT are successful in achieving 100% accuracy, while other timing methods are less accurate. Between these two timing methods that achieved the best accuracy, CIFT has higher total hits than the cumulative energy method, which indicates the positions computed with CIFT plus table lookup are more closely clustered around the known PD points.

To further quantify the performance of these methods, the next set of quality measures are computed with a smaller error range of 300 mm, which is just large enough to enclose the antennas that triggered the sample set of PD events. With a tighter error bound, the results in Table 3 indicate that the CIFT has better localization accuracy than the cumulative energy method. The accuracy results for PD2 are much lower, since the noisiest channel for the PD2 data has a much lower SNR than the noisiest channel for PD1. Figure 13 and Figure 14 show the 300 mm localization results of cumulative energy and the CIFT, respectively. The bounding box for CIFT is weighted more towards the cluster of actual PDs, since more of the localized PD sources are near the actual PDs than in the cumulative energy case.

### 5.3. Additional Validation Tests with Arbitrary PD Locations

Since the lab-triggered PD events are limited to very specific locations, the next set of tests uses simulation to create PD events in all possible locations inside the insulation of two different transformers named DL23 and TL19. A small amount of Gaussian noise is added to the FDTD simulation data to challenge timing methods in this set of tests.

To provide direct evidence of how effective are different timing methods, Figure 10 shows the signal arrival time computed by three different methods. The blue lines represent the signal, which begins at 5 nanoseconds. CIFT consistently provides the timing closest to the actual signal arrival, which shows that the CIFT method indeed offers better timing results.

The tests with simulated PD events are generated from each mesh point of the FDTD mesh, where the two neighboring mesh points are 300 mm apart. Altogether, 8750 simulated PD events are generated from the DL23 model. The proposed method of CIFT + FDTD table lookup produced more accurate locations and the average error from all starting points is about 340 mm in Euclidean distance, which is fairly close to the size of the FDTD mesh of 300 mm.

The distribution of errors in the computed locations from DL23 tests are shown in Figure 15. The prediction errors from the CIFT method are a lot lower than those of the cumulative energy method. In all four cases, there are derived PD sources that are far from their actual locations. However, based on the range of the horizontal axes, we see the maximum error is smaller with combination of CIFT and FDTD table lookup. We achieved similar results from the TL19 transformer.

The synthetic test cases with DL23 and TL19 are more challenging than the actual PDs from KE20 because both DL23 and TL19 are more complex than DE20 in their internal structures, and because the synthetic test cases include all possible locations inside the transformers. Even though our proposed combination of CIFT for timing and table lookup for localization is able to predict the location more accurately than the best of the known methods, we believe it is worthwhile to investigate the worst-case scenarios shown in Figure 15.

## 6. Conclusions

The goal of this data analysis work is to accurately locate the source of partial discharges in a transformer. This task is broken into two steps: compute the arrival time at each sensor and then locate the source based on the time differences. Based on a study of known methods to determine the arrival time from sensor measurements, a new composite method is proposed. This composite method named CIFT first employs a convolutional filter to determine a time window containing the arrival time and then uses a moving average to determine the precise arrival time.

To compute the location from the arrival time, the commonly used algorithm is multilateration, which assumes the PD signals travel in straight lines toward the sensors at a constant known speed. However, this assumption is not true for large transformers with complex internal metal structures. To accommodate the complex internal structures, a reference table of arrival time differences is generated from a regular discretization of an insulator inside a transformer. Given a set of arrival times at four sensors, one could lookup the mesh cell with the best matching time differences from the reference table. We construct this reference table with FDTD simulations using the structures of actual transformers.

The combination of CIFT and the table lookup method proved to be very effective on real PD signals induced in the lab. In determining the arrival time, CIFT can generally locate the time of arrival that is earlier than other methods, including the cumulative energy method and the energy criteria. In terms of location accuracy, the table lookup method is able to locate the PD to within 300 mm much more frequently than other methods. Our method consistently achieved more total hits than the existing cumulative energy method. With a tolerance of 300 mm, in the high SNR events, the accuracy with cumulative energy method is 91% and our method is 95%. On the low SNR events, the accuracy increased from 13% to 48%, a 3.7X improvement.

For many preventive actions, the location accuracy of 300 mm is sufficiently useful. However, there are a few cases in Figure 15 where the errors are more than a meter. It would be necessary to understand the root causes of these cases. In addition, there are a few other questions that might deserve additional investigation. For example, the implementation of CIFT method involves several parameters that could use more fine tuning. In addition, errors from FDTD table lookup might depend on the FDTD mesh resolution; it would be useful to increase the mesh resolution to determine if it is possible to improve the location accuracy.

## 7. Patents

U.S. Patent 10,908,972 B2, “Methods, systems, and devices for accurate signal timing of power component events”, 2021 [36].

## Figures and Tables

**Figure 1 sensors-23-01789-f001:**
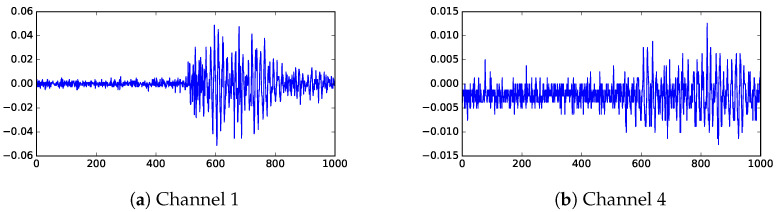
Sample raw data from UHF. The horizontal axis is time steps (where each step is 0.4 ns) and the vertical axis shows UHF measurements (in V).

**Figure 2 sensors-23-01789-f002:**
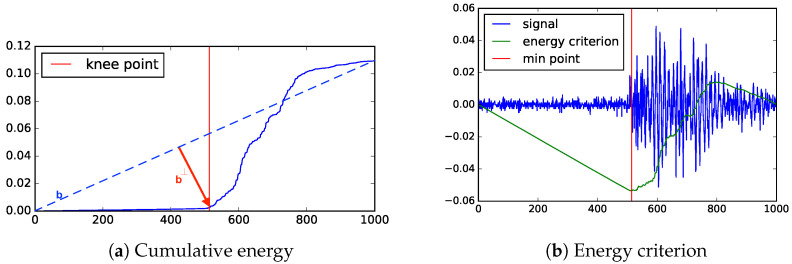
Two state-of-the-art timing methods. The horizontal axis is time steps and the vertical axis is energy (in V${}^{2}$).

**Figure 3 sensors-23-01789-f003:**
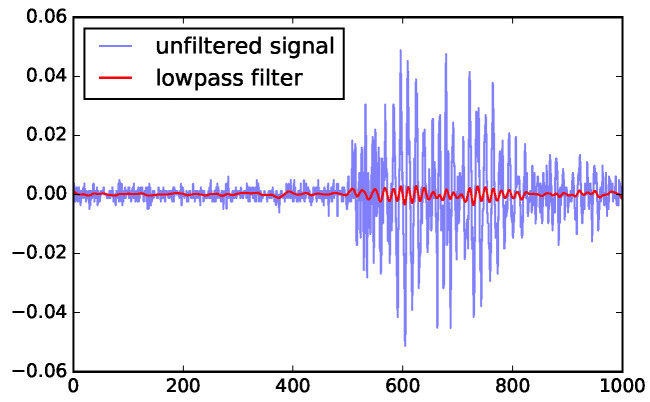
Low-pass filter could remove too much of the signal, an illustration with a series of high SNR (channel 1) data. The horizontal axis is time steps (where each step is 0.4 ns) and the vertical axis shows UHF measurements before and after filtering (in V).

**Figure 4 sensors-23-01789-f004:**
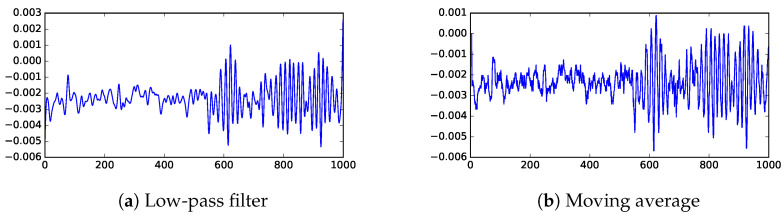
Filtered version of channel 4 data (low SNR). These two popular filtering techniques produce similar range of values. The horizontal axis is time steps (where each step is 0.4 ns) and the vertical axis shows UHF measurements processed with different techniques (in V).

**Figure 5 sensors-23-01789-f005:**
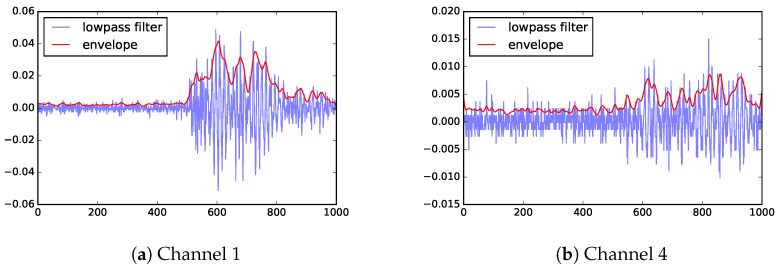
Envelope of two sample data sets. The horizontal axis is time steps (where each step is 0.4 ns) and the vertical axis shows UHF measurements (blue) along with envelope (red) (in V).

**Figure 6 sensors-23-01789-f006:**
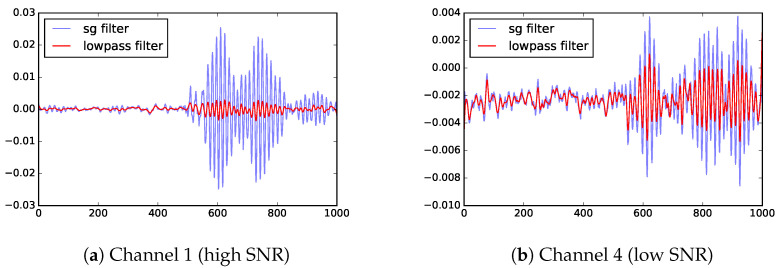
Savitzky–Golay (SG) filter works considerably better than low-pass filter. The horizontal axis is time steps (where each step is 0.4 ns) and the vertical axis shows UHF measurements processed with different techniques (in V).

**Figure 7 sensors-23-01789-f007:**
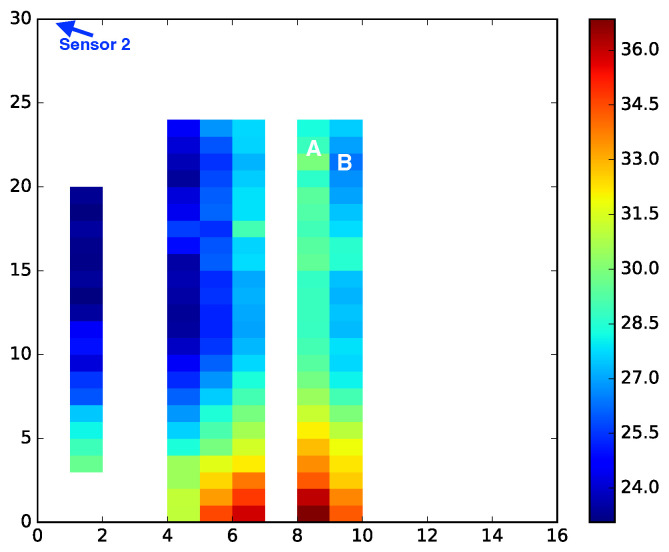
A heat map of arrival time to sensor 2 (located at the upper left corner) from a cross section of the FDTD mesh for KE20. The two axes are given in FDTD indices. Note that FDTD cell A is closer to sensor 2 than cell B, but the travel time from cell A is longer (green) than from cell B (blue).

**Figure 8 sensors-23-01789-f008:**
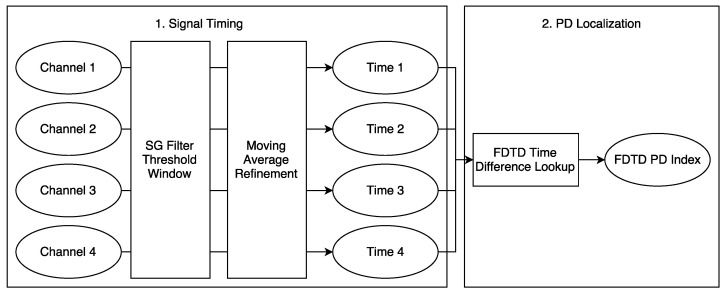
An overview of proposed PD localization procedure.

**Figure 9 sensors-23-01789-f009:**
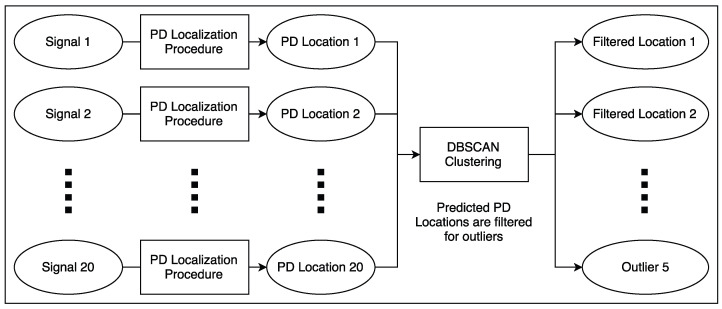
Procedure for finding outliers.

**Figure 10 sensors-23-01789-f010:**
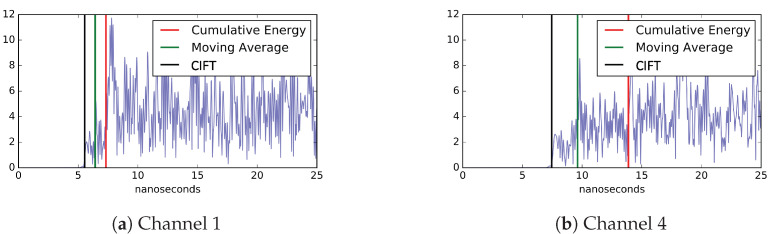
Signal arrival time of a simulated PD event in DL23. The horizontal axis is time and the vertical axis is the measured voltage (from FDTD simulation). In all cases, CIFT is able to identify the signal arrival time close to the known start time of 5 ns. In a low SNR case, shown in (**b**), the arrival time computed by the cumulative energy method is nearly 10 ns later than the actual signal start time, while the error of CIFT is only about 3 ns.

**Figure 11 sensors-23-01789-f011:**
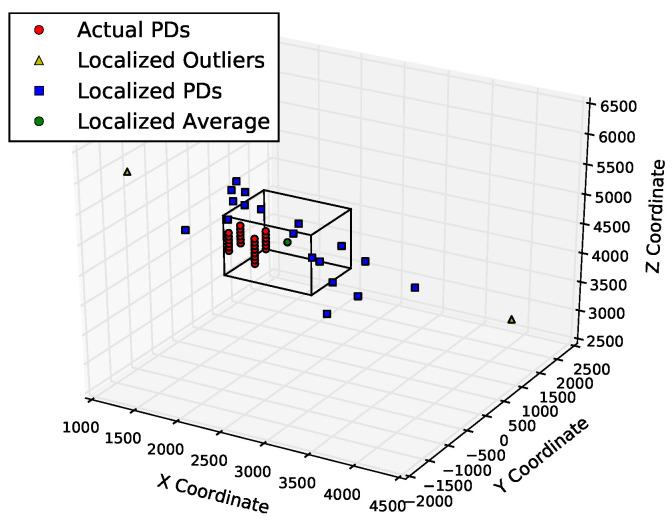
Localization results of multilateration on PD1 in KE20. The three axes are three dimensions of space (measured in mm).

**Figure 12 sensors-23-01789-f012:**
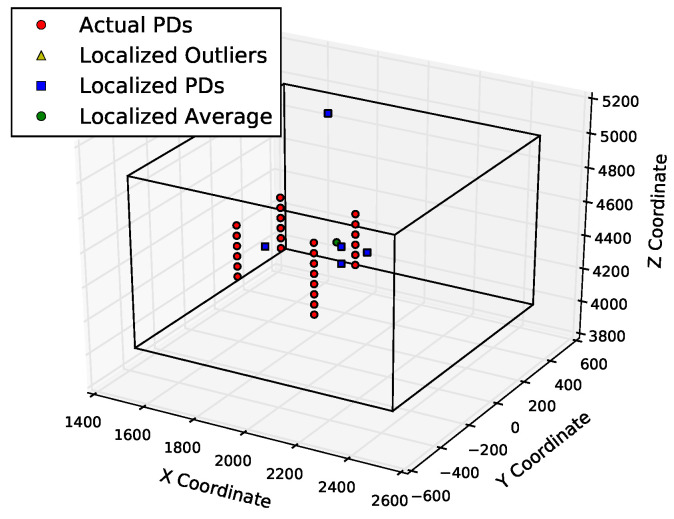
Localization results from FDTD table lookup on PD1 in KE20. The three axes are three dimensions of space (measured in mm). Many PD signals are determined to be from the same location, and there is no obvious outliers. All computed PD locations are well within the bounding box defined by the 500 mm tolerance.

**Figure 13 sensors-23-01789-f013:**
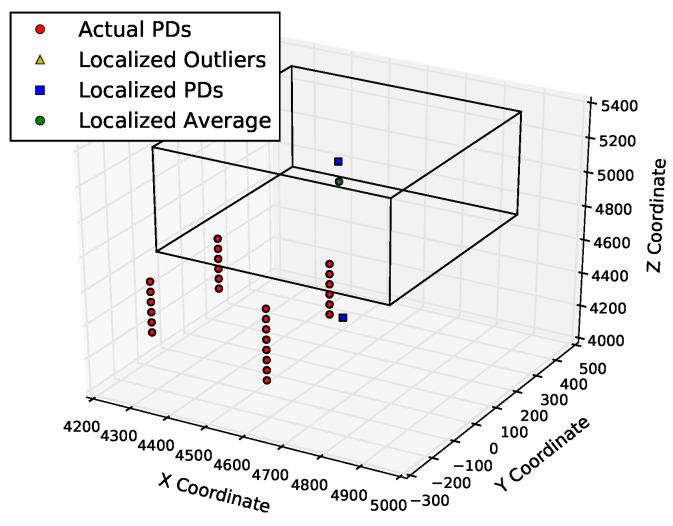
Localization results of cumulative energy are usually more than 300 mm away from the actual PD2 locations. The three axes are three dimensions of space (in mm).

**Figure 14 sensors-23-01789-f014:**
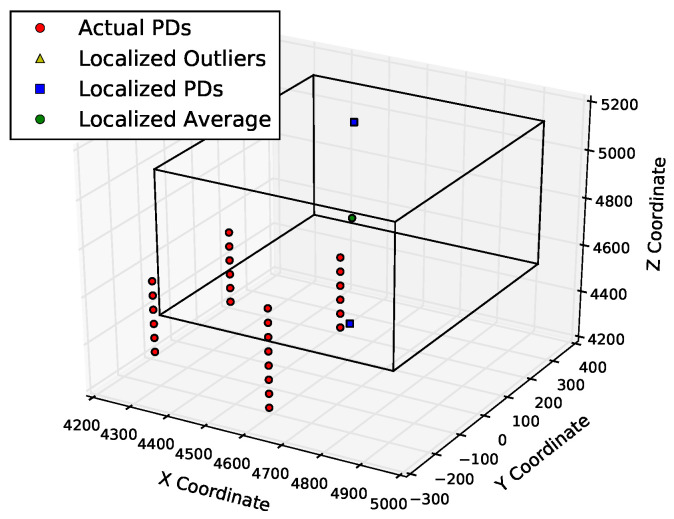
Localization results of CIFT are largely within 300 mm from the actual PD2 locations. The three axes are three dimensions of space (in mm).

**Figure 15 sensors-23-01789-f015:**
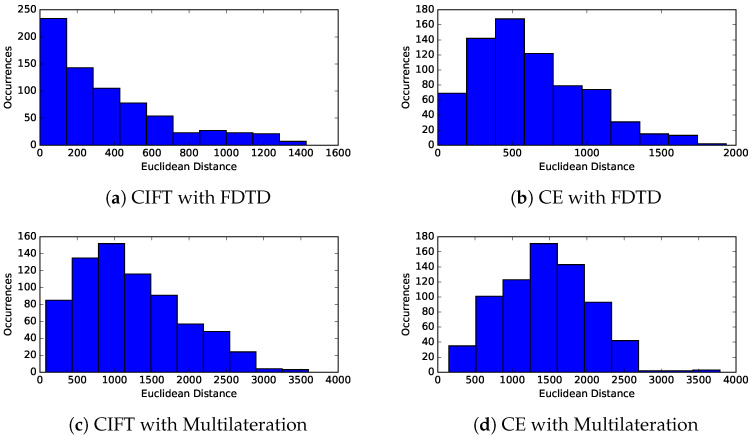
Error distribution (histogram) of PD events from DL23. The horizontal axis is the computed error. Each occurance is from one simulated PD event from a FDTD cell. The average error using CIFT with FDTD is about 340 mm, which is close the FDTD mesh size of 300 mm and smaller than other methods tested. All methods have some cases with large errors that require further investigation.

**Table 1 sensors-23-01789-t001:** Results with different localization methods and 500 mm tolerance.

Localization Method	PD	Outliers	Accuracy	Total Hits
Multilateration	PD1	2	0.4	69
	PD2	9	0.79	196
Table lookup	PD1	0	1	536
	PD2	0	1	238

**Table 2 sensors-23-01789-t002:** Localization (table lookup) results with different timing methods and 500mm tolerance.

Timing Method	PD	Outliers	Accuracy	Total Hits
Noise Cancellation	PD1	15	0	52
	PD2	15	0	30
Wavelength Comparison	PD1	13	0	0
	PD2	12	1	142
Data Spread	PD1	4	0.22	104
	PD2	13	1	260
Moving Average	PD1	13	1	206
	PD2	0	1	216
Threshold w corr	PD1	10	0.42	84
	PD2	2	1	376
Envelope	PD1	10	0.92	286
	PD2	0	0	0
Cumulative Energy	PD1	0	1	536
	PD2	0	1	238
CIFT	PD1	0	1	554
	PD2	0	1	382

**Table 3 sensors-23-01789-t003:** Localization (table lookup) results with different timing methods and 300 mm tolerance.

Timing Method	PD	Outliers	Accuracy	Total Hits
Cumulative Energy	PD1	0	0.91	292
	PD2	0	0.13	42
CIFT	PD1	0	0.95	298
	PD2	0	0.48	154

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
