# Peer review of "Locating Partial Discharges in Power Transformers with Convolutional Iterative Filteringâ€"

_sensors, 2023, doi:10.3390/s23041789_

Round 1
Reviewer 1 Report
1. Complete the state-of-the-art is incomplete
2. Add information about sampling rates
3. Highlight the main contributions according the limitation of other similar works
4. Improve Figure 14
5. Use number of equations
Author Response
Thanks for the suggestion. We have reviewed the English writing. Please find attached the point-to-point responses to reviewer's comments.

Reviewer 2 Report
The paper presents the study results about study of locating partial discharges in power transformers with convolutional iterative filtering. Authors present algorithms which may help to locate partial discharges event using data from ultra high frequency sensors inside the transformer. They evaluate effectiveness of these algorithms which were tested with a set of laboratory-triggered PD events and two sets of simulated PD events inside transfers in production use. Obtained results proved that new approach provides more accurate locations than the best-known data analysis algorithms, and the difference is particularly large.
Dear authors, thank you very much for interesting paper, which presents new method in order to find a source of partial discharges in power transformers. I put some comments and questions.
Comments:
1. Introduction, even if it is short, is well recognized. Authors indicate main problem, defects which may occur in power transformers, and negative activity of partial discharges. They describe methods of measurements of partial discharges, where important role of ultra high frequency method is indicated.
2. Fig.1. – I think, all axis should be described on the figure. Please complete the figure and rest of all.
3. Next chapter presents related investigations to the paper topic. Authors used many references to describe state of investigation of UHF method, used in partial discharge detection in power transformers.
4. Fig.2. – the same comments, please complete description of X and Y axis. It is difficult to understand what relationships the figures are presenting.
5. If some formula is presented, please put information about source of the formula – reference – and mark it as formula, means for example (1). Please complete.
6. Please describe the elements of formula (1) and put information about units of the elements.
7. Low-pass filter. Explain please how the filter was designed. What frequency is transmitted and what blocked by the filter. Please consider some more technical information about the filter.
8. Noise cancelation – the same comments like in case of 7.
9. CIFT – please explain the word.
10. Fig.15. Concurrency – how do authors define the parameter?
11. Main conclusions – the topic and presented results are very interesting. Anyway, there are a lot of points to be explained, such as description of all figures (axis), formulas sources, definition of some parameters.
Author Response
Thanks for the careful review and constructive suggestions. We have updated the document to address the suggestions. Please find attached the point-to-point responses to reviewer's comments.
